# CORRECTING MOMENTUM IN TEMPORAL DIFFERENCE LEARNING

## ABSTRACT

A common optimization tool used in deep reinforcement learning is momentum, which consists in accumulating and discounting past gradients, reapplying them at each iteration. We argue that, unlike in supervised learning, momentum in Temporal Difference (TD) learning accumulates gradients that become doubly stale: not only does the gradient of the loss change due to parameter updates, the loss itself changes due to bootstrapping. We first show that this phenomenon exists, and then propose a first-order correction term to momentum. We show that this correction term improves sample efficiency in policy evaluation by correcting target value drift. An important insight of this work is that deep RL methods are not always best served by directly importing techniques from the supervised setting.

## 1 INTRODUCTION

Temporal Difference (TD) learning (Sutton, 1988) is a fundamental method of Reinforcement Learning (RL), which works by using estimates of future predictions to learn to make predictions about the present. To scale to large problems, TD has been used in conjunction with deep neural networks (DNNs) to achieve impressive performance (e.g. Mnih et al., 2013; Schulman et al., 2017; Hessel et al., 2018). Unfortunately, the naive application of TD to DNNs has been shown to be brittle (Machado et al., 2018; Farebrother et al., 2018; Packer et al., 2018; Witty et al., 2018), with extensions such as the $n$-step TD update (Fedus et al., 2020) or the TD($\lambda$) update (Molchanov et al., 2016) only marginally improving performance and generalization capabilities when coupled with DNNs.

Part of the success of DNNs, including when applied to TD learning, is the use of adaptive or accelerated optimization methods (Hinton et al., 2012; Sutskever et al., 2013; Kingma & Ba, 2015) to find good parameters. In this work we investigate and extend the momentum algorithm (Polyak, 1964) as applied to TD learning in DNNs. While accelerated TD methods have received some attention in the literature, this is typically done in the context of linear function approximators (Baxter & Bartlett, 2001; Meyer et al., 2014; Pan et al., 2016; Gupta et al., 2019; Gupta, 2020; Sun et al., 2020), and while some studies have considered the mix of DNNs and TD (Zhang et al., 2019; Romoff et al., 2020), many are limited to a high-level analysis of hyperparameter choices for *existing* optimization methods (Sarigül & Avci, 2018; Andrychowicz et al., 2020); or indeed the latter are simply applied as-is to train RL agents (Mnih et al., 2013; Hessel et al., 2018). For an extended discussion of related work, we refer the reader to appendix A.

As a first step in going beyond the naive use of supervised learning tools in RL, we examine momentum. We argue that momentum, especially as it is used in conjunction with TD and DNNs, adds an additional form of bias which can be understood as the staleness of accumulated information. We quantify this bias, and propose a corrected momentum algorithm that reduces this staleness and is capable of improving performance.

### 1.1 REINFORCEMENT LEARNING AND TEMPORAL DIFFERENCE LEARNING

A Markov Decision Process (MDP) (Bellman, 1957; Sutton & Barto, 2018) $\mathcal{M} = \langle S, A, R, P, \gamma \rangle$ consists of a state space $S$, an action space $A$, a reward function $R : S \to \mathbb{R}$ and a transition function $P(s'|s, a)$. RL agents usually aim to optimize the expectation of the long-term return, $G(S_t) = \sum_{k=t}^{\infty} \gamma^{k-t} R(S_k)$ where $\gamma \in [0, 1)$ is called the discount factor. Policies $\pi(a|s)$ map states to action distributions. Value functions $V^\pi$ and $Q^\pi$ map states and states-action pairs to

expected returns, and can be written recursively:

$$V^\pi(S_t) = \mathbb{E}_\pi[G(S_t)] = \mathbb{E}_\pi[R(S_t, A_t) + \gamma V(S_{t+1})|A_t \sim \pi(S_t)]$$
$$Q^\pi(S_t, A_t) = \mathbb{E}_\pi[R(S_t, A_t) + \gamma\sum_a \pi(a|S_{t+1})Q(S_{t+1}, a)]$$

We approximate $V^\pi$ with $V_\theta$. We can train $V_\theta$ via regression to observed values of $G$, but these recursive equations also give rise to the *Temporal Difference (TD)* update rules for policy evaluation, relying on current estimates of $V$ to *bootstrap*, which for example in the *tabular* case is written as:

$$V(S_t) \leftarrow V(S_t) - \alpha(V(S_t) - (R(S_t) + \gamma V(S_{t+1}))), \tag{1}$$

where $\alpha \in [0, 1)$ is the step size. Alternatively, estimates of this update can be performed by a so-called *semi-gradient* (Sutton & Barto, 2018) algorithm where the "TD(0) loss" is minimized:

$$\theta_{t+1} = \theta_t - \alpha\nabla_{\theta_t}\left(V_{\theta_t}(S_t) - (R(S_t) + \gamma\bar{V}_{\theta_t}(S_{t+1}))\right)^2, \tag{2}$$

with $\bar{V}$ meaning we consider $V$ constant for the purpose of gradient computation.

## 1.2 BIAS AND STALENESS IN MOMENTUM

The usual form of momentum (Polyak, 1964; Sutskever et al., 2013) in stochastic gradient descent (SGD) maintains an exponential moving average with factor $\beta$ of gradients w.r.t. to some objective $J$, changing parameters $\theta_t \in \mathbb{R}^n$ ($t$ is here SGD time rather than MDP time) with learning rate $\alpha$:

$$\mu_t = \beta\mu_{t-1} + (1 - \beta)\nabla_{\theta_{t-1}}J_t(\theta_{t-1}) \tag{3}$$
$$\theta_t = \theta_{t-1} - \alpha\mu_t \tag{4}$$

We assume here that the objective $J_t$ is time-dependent, as is the case for example in minibatch training or online learning. Note that other similar forms of this update exist, notably Nesterov's accelerated gradient method (Nesterov, 1983), as well as undampened methods that omit $(1 - \beta)$ in (3) or replace $(1 - \beta)$ with $\alpha$, found in popular deep learning packages (Paszke et al., 2019).

We make the observation that, at time $t$, the gradients accumulated in $\mu$ are *stale*. They were computed using past parameters rather than $\theta_t$, and in general we'd expect $\nabla_{\theta_t}J_t(\theta_t) \neq \nabla_{\theta_k}J_t(\theta_k)$, $k < t$. As such, the update in (4) is a biased update.

In supervised learning where one learns a mapping from $x$ to $y$, this staleness only has one source: $\theta$ changes but the target $y$ stays constant. We argue that in TD learning, momentum becomes **doubly** stale: not only does the value network change, but the target (the equivalent of $y$) itself changes[1] with every parameter update. Consider the TD objective in (2), when $\theta$ changes, not only does $V(s)$ change, but $V(s')$ as well. The objective itself changes, making past gradients stale and less aligned with recent gradients (even more so when there is gradient interference (Liu et al., 2019; Achiam et al., 2019; Bengio et al., 2020), constructive *or* destructive).

Note that several sources of bias already exist in TD learning, notably the traditional parametric bias (of the bias-variance tradeoff when selecting capacity), as well as the bootstrapping bias (of the error in $V(s')$ when using it as a target; using a frozen target prevents this bias from compounding). We argue that the staleness in momentum we describe is an additional form of bias, slowing down or preventing convergence. This has been hinted at before, e.g. Gupta (2020) suggests that momentum hinders learning in linear TD(0).

We wish to understand and possibly correct this staleness in momentum. In this work we propose answers to the following questions:

- Is this bias significant in **supervised learning**? **No**, the bias exists but has a minimal effect at best when comparing to an unbiased oracle.
- Is this bias significant in **TD learning**? **Yes**, we can quantify the bias, and comparisons to an unbiased oracle reveal significant differences.
- Can we **correct** $\mu_t$ to remove this bias? **Yes**, we derive an online update that approximately corrects $\mu_t$ using only first order derivatives.
- Does the correction help in TD learning? **Yes**, using a staleness-corrected momentum **improves sample complexity**, in **policy evaluation**, especially in an online setting.

---

[1]Interestingly, even in most recent value-based control works (Hessel et al., 2018) a (usually *frozen*) copy is used for *stability*, meaning that the target only changes when the copy is updated. This is considered a "trick" which it would be compelling to get rid of, since it slows down learning, and since most recent policy-gradient methods (which still use a value function) do not make use of such copies (Schulman et al., 2017).

## 2    CORRECTING MOMENTUM

### 2.1    IDENTIFYING BIAS

Here we propose an experimental protocol to quantify the bias of momentum. We assume we are minimizing some objective in the minibatch or online setting. In such a stochastic setting, momentum is usually understood as a variance reduction method, or as approximating the large-batch setting (Botev et al., 2017), but as discussed above, momentum also induces bias in the optimization.

We note that $\mu_t$, the momentum at time $t$, can be rewritten as:

$$\mu_t = (1 - \beta) \sum_{i=1}^{t} \beta^{t-i} \nabla_{\theta_i} J_i(\theta_i), \tag{5}$$

and argue that an ideal *unbiased* momentum $\mu_t^*$ would approximate the large batch case by only discounting past minibatches and using current parameters rather than past parameters:

$$\mu_t^* \stackrel{\text{def}}{=} (1 - \beta) \sum_{i=1}^{t} \beta^{t-i} \nabla_{\theta_t} J_i(\theta_t). \tag{6}$$

Note that the only difference between (5) and (6) is the use of $\theta_i$ vs $\theta_t$. The only way to *exactly* compute $\mu_t^*$ is to recompute the entire sum after every parameter update. We will consider this our **unbiased oracle**. To compute $\mu_t^*$ empirically, since beyond a certain power $\beta^k$ becomes small, we will use an effective horizon of $h = 2/(1 - \beta)$ steps (i.e. start at $i = t - h$ rather than at $i = 1$).

We call the difference between $\mu_t$ and $\mu_t^*$ the *bias*, but note that in the overparameterized stochastic gradient case, minor differences in gradients can quickly send parameters in different regions. This makes the direct measure of $\|\mu_t - \mu_t^*\|$ uninformative. Instead, we measure the **optimization bias** by simply comparing the loss of a model trained with $\mu_t$ against that of a model trained with $\mu_t^*$.

Finally, we note that, in RL, momentum approximating the batch case is related to (approximately) replaying an entire buffer at once (instead of sampling transitions). This has been shown to also have interesting forms in the linear case (van Seijen & Sutton, 2015), reminiscent of the correction derived below. We also note that, while the mathematical expression of momentum and eligibility traces (Sutton, 1988) *look* fairly similar, they estimate a very different quantity (see appendix A).

### 2.2    CORRECTING BIAS

Here we propose a way to approximate $\mu^*$, and so derive an approximate correction to the bias in momentum for supervised learning, as well as for Temporal Difference (TD) learning.

We consider here the simple regression and TD(0) cases, for the online (minibatch size 1) case. We show the full derivations, and results for any minibatch size, TD($\lambda$) and n-step TD, in appendix B. In least squares regression with loss $\delta^2$ we can write the gradient $g_t$ as:

$$g_t(\theta_t) = (y_t - f_{\theta_t}(x_t))\nabla_{\theta_t} f_{\theta_t}(x_t) = \nabla_{\theta_t} \delta_t^2 / 2. \tag{7}$$

We would like to "correct" this gradient as we move away from $\theta_t$. A simple way to do so is to compute the Taylor expansion around $\theta_t$ of $g_t$:

$$g_t(\theta_t + \Delta\theta) = g_t(\theta_t) + \nabla_{\theta_t} g_t(\theta_t)^\top \Delta\theta + o(\|\Delta\theta\|_2^2), \tag{8}$$

$$\approx g_t(\theta_t) + (\delta\nabla_{\theta_t}^2 f_{\theta_t}(x_t) - \nabla_{\theta_t} f_{\theta_t}(x_t) \otimes \nabla_{\theta_t} f_{\theta_t}(x_t))^\top \Delta\theta, \tag{9}$$

where $\otimes$ is the outer product, $\nabla^2$ the second derivative. We note that the term multiplying $\Delta\theta$ is commonly known as "the Hessian" of the loss. We also note that Taylor expansions around parameters have a rich history in deep learning (LeCun et al., 1990; Molchanov et al., 2016), and that in spite of its non-linearity, the parameter space of a deep network is filled with locally consistent regions (Balduzzi et al., 2017) in which Taylor expansions are accurate.

The same correction computation can be made for TD(0) with TD loss $\delta^2$. Remember that we write $t$ as the learning time; we denote an MDP transition as $(s_t, a_t, r_t, s_t')$, making no assumption on the distribution of transitions:

$$g_t(\theta_t) = (V_{\theta_t}(s_t) - r_t - \gamma V_{\theta_t}(s_t'))\nabla_{\theta_t} V_{\theta_t}(s_t) = \nabla_{\theta_t} \delta_t^2 / 2, \tag{10}$$

$$g_t(\theta_t + \Delta\theta) \approx g_t(\theta_t) + (\nabla_{\theta_t}(V_{\theta_t}(s_t) - \gamma V_{\theta_t}(s_t')) \otimes \nabla_{\theta_t} V_{\theta_t}(s_t) + \delta\nabla_{\theta_t}^2 V_{\theta_t}(s_t))^\top \Delta\theta. \tag{11}$$

Here, because we are using a semi-gradient method, the term multiplying $\Delta\theta$ is not exactly the Hessian: when computing $\nabla_\theta \delta^2$, we hold $V(s')$ constant, but when computing $\nabla_\theta g$, we need to consider $V_\theta(s')$ a function of $\theta$ as it affects $g$, and so compute its gradient.[2]

Without loss of generality, let us write equations like (9) and (11) using the matrix $Z_t \in \mathbb{R}^{n \times n}$:

$$g_t(\theta_t + \Delta\theta) \approx g_t(\theta_t) + Z_t^\top \Delta\theta, \qquad (12)$$

where the form of $Z_t$, the "Taylor term", depends on the loss (e.g. the Hessian in (9)).

Recall that in (6) we define $\mu^*$ as the discounted sum of gradients using $\theta_t$ for losses $J_i$, $i \leq t$. We call those gradients *corrected* gradients, $g_i^t := g_i(\theta_i + (\theta_t - \theta_i))$. At timestep $t$ we update $\theta_t$ with $\alpha\mu_t$, thus we substitute $\Delta\theta = \theta_t - \theta_i = -\alpha\sum_{k=i}^{t} \mu_k$ in (12) and get:

$$\hat{g}_i^t \stackrel{\text{def}}{=} g_i(\theta_i) - \alpha Z_i^\top \sum_{k=i}^{t-1} \mu_k \approx g_i^t. \qquad (13)$$

We can now approximate the unbiased momentum $\mu_t^*$ using (13), which we denote $\hat{\mu}$:

$$\hat{\mu}_t \stackrel{\text{def}}{=} (1 - \beta) \sum_{i=1}^{t} \beta^{t-i} \hat{g}_i^t \qquad (14)$$

$$= \mu_t - \alpha(1 - \beta) \sum_{k=1}^{t-1} \sum_{i=1}^{k} \beta^{t-i} Z_i^\top \hat{\mu}_k. \qquad (15)$$

Noting that $\mu_t$ can be computed as usual and that the second term of (15) has a recursive form, which we denote $\eta_t$ (see appendix B), we rewrite $\hat{\mu}$ as follows:

$$\hat{\mu}_t = \mu_t - \eta_t \qquad (16)$$

$$\eta_t = \beta\eta_{t-1} + \alpha\beta\zeta_{t-1}^\top \hat{\mu}_{t-1} \qquad \text{(correction term)} \qquad (17)$$

$$\mu_t = (1 - \beta) \sum_{i=1}^{t} \beta^{t-i} g_i = (1 - \beta)g_t + \beta\mu_{t-1} \qquad \text{(normal momentum)} \qquad (18)$$

$$\zeta_t = (1 - \beta) \sum_{i=1}^{t} \beta^{t-i} Z_i = (1 - \beta)Z_t + \beta\zeta_{t-1}. \qquad \text{("momentum" of Taylor terms)} \qquad (19)$$

Algorithmically, this requires maintaining 2 vectors $\mu$ and $\eta$ of size $n$, the number of parameters, and one matrix $\zeta$ of size $n \times n$. In practice, we find that only maintaining the diagonal of $\zeta$ can also work and can avoid the quadratic growth in $n$.

The computation of $Z$ also calls on computing second order derivatives $\nabla^2 f$ (e.g. in (9) and (11)), which is impractical for large architectures. In this work, as is commonly done due to the usually small magnitude of $\nabla^2 f$ (Bishop, 2006, section 5.4), we ignore them and only rely on the outer product of gradients.

Ignoring the second derivatives, computing $Z$ for TD(0) requires 3 backward passes, for $g = \nabla\delta^2$, $\nabla\gamma V(s')$, and $\nabla V(s)$. In the online case $g = \delta\nabla V(s)$, requiring only 2 backward passes, but in the more general minibatch of size $m > 2$ case, it is more efficient with modern automatic differentiation packages to do 3 backward passes than $2m$ passes (see appendix B.4).

Finally, we note that this method easily extends to forward-view TD($\lambda$) and $n$-step TD methods, all that is needed is to compute $Z$ appropriately (see appendix B.2).

## 3 EXPERIMENTAL RESULTS

We evaluate our proposed correction as well as the oracle (the best our method could perform) on common RL benchmarks and supervised problems. We evaluate each method with a variety of

---

[2]This computation of the gradient of $V_\theta(s')$ may remind the reader of the so-called *full gradient* or *residual gradient* (Baird, 1995), but its purpose here is very different: we care about learning using semi-gradients, TD(0) is a principled algorithm, but we also care about *how* this semi-gradient evolves as parameters change, and thus we need to compute $\nabla_\theta V_\theta(s')$.

hyperparameters and multiple seeds for each hyperparameter setting. The full range of hyperparameters used as well as architectural details can be found in appendix D.

We will use the following notation for the optimizer used to train models: $\mu$ is the usual momentum, i.e. (3)&(4), and serves as a baseline. $\mu^*$ is our oracle, defined in (6). $\hat{\mu}$ is the proposed correction, with updates as in (16)-(19), using the outer product approximation of $Z$. $\hat{\mu}_{\text{diag}}$ is the proposed correction, but with a diagonal approximation of $Z$. Throughout our figures, shaded areas are bootstrapped 95% confidence intervals over hyperparameter settings (if applicable) and runs.

## 3.1 SUPERVISED LEARNING

We first test our hypothesis, that there is bias in momentum, in a simple regression task and SVHN. For regression, we task a 4-layer MLP to regress to a 1-d function from a 1-d input. For illustration we use a smooth but non-trivial mixture of sines of increasing frequency in the domain $x \in [-1, 1]$:

$$y(x) = 0.5\sin(2.14(x+2)) + 0.82\sin(9x+0.4) + \\ 0.38\sin(12x) + 0.32\sin(38x - 0.1) \quad (20)$$

Note that this choice is purely illustrative and that our findings extend to similar simple functions. We train the model on a fixed sample of 10k uniformly sampled points. We measure the mean squared error (MSE) when training a small MLP with momentum SGD versus our oracle momentum $\mu^*$ and the outer product correction $\hat{\mu}$. As shown in Figure 1, we find that while there is a significant difference between the oracle and the baseline ($p \approx 0.001$ from Welsh's t-test), the difference is fairly small, and is no longer significant when using the corrected $\hat{\mu}$ ($p \approx 0.1$).

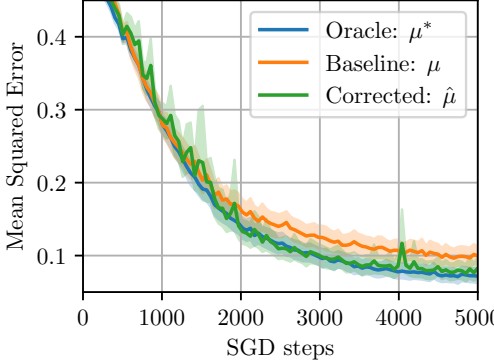
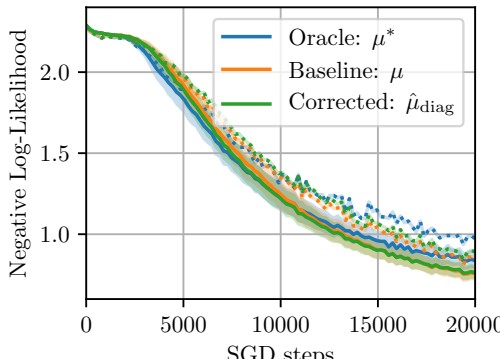

Figure 1: Regression to (20) with varying momentums (10 seeds per setting). Note that the difference between $\mu$ and $\hat{\mu}$ is not significant, but $\mu$ and $\mu^*$ is.

Figure 2: Classification on SVHN with varying momentums (5 seeds per setting). Dotted lines are test losses. The only significant difference is between the training loss of $\mu^*$ and $\mu$.

We compare training a small convolutional neural network on SVHN (Netzer et al., 2011) with momentum SGD versus our oracle momentum $\mu^*$ and the diagonalized correction momentum $\hat{\mu}_{\text{diag}}$. The results are shown in Figure 2. We do not find significant differences except between the training loss of $\mu^*$ and $\mu$, and find that the oracle performs *worse* than the normal and corrected momentum.

From these experiments we conclude that, in supervised learning, there exists a quantifiable optimization bias to momentum, but that correcting it does not appear to offer any benefit. It improves performance only marginally at best, and degrades it at worst. This is consistent with the insight that $\mu^*$ approximates the large batch gradient, and that large batches are often associated with overfitting or poor convergence in supervised learning (Wilson & Martinez, 2003; Keskar et al., 2016).

## 3.2 TEMPORAL DIFFERENCE LEARNING

We now test our hypotheses, that there is optimisation bias and that we can correct it, on RL problems. First, we test policy evaluation of the optimal policy on the Mountain Car problem (Singh

& Sutton, 1996) with a small MLP. We also test Acrobot and Cartpole environments and find very similar results (see appendix C). We then test our method on Atari (Bellemare et al., 2013) with convolutional networks.

Figure 3 shows policy evaluation on Mountain Car using a replay buffer (on-policy state transitions are sampled i.i.d. in minibatches). We compare the loss distributions (across hyperparameters and seeds) at step 5k, and find that all methods are significantly different ($p < 0.001$) from one another. Figure 4 shows online policy evaluation, i.e. the transitions are generated and learned from once, in-order, and one at a time (minibatch size of 1). There we see that the oracle $\mu^*$ and full corrected version $\hat{\mu}$ are significantly different from the baseline $\mu$ ($p < 0.001$) and diagonalized correction $\hat{\mu}_{\text{diag}}$, as well as $\mu$ from $\hat{\mu}_{\text{diag}}$, while $\mu^*$ and $\hat{\mu}$ are not significantly different ($p > 0.1$).

This suggests that the $\zeta$ matrix carries useful off-diagonal temporal information about parameters which co-vary, especially when the data is not used uniformly during learning. We test another possible explanation, which is that performance is degraded in online learning because the batch size is 1 (rather than 16 or 32 as in Figure 3). We find that a batch size of 1 does degrade $\hat{\mu}_{\text{diag}}$'s performance significantly, but does not fully explain its poor online performance (see appendix Figure 10).

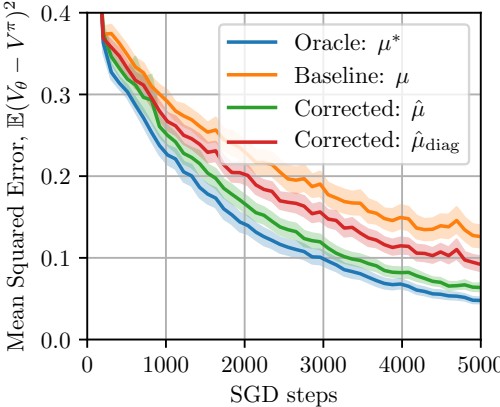 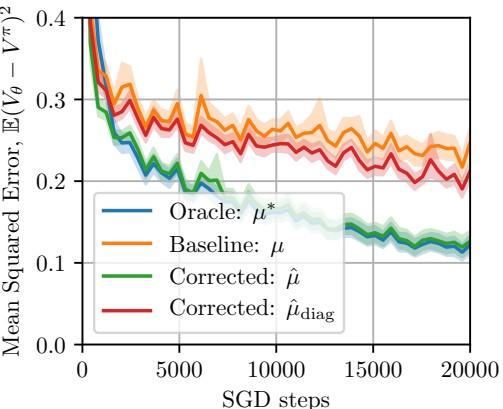

Figure 3: TD(0) policy evaluation on Mountain Car with varying momentums on a **replay buffer**. The MSE is measured against a pretrained $V^\pi$ (10 seeds per setting). At step 5k, all methods are significantly different.

Figure 4: TD(0) **online** policy evaluation on Mountain Car, transitions are seen in-order. The MSE is measured against a pretrained $V^\pi$ (50 seeds per setting). At step 20k, $\mu^*$ and $\hat{\mu}$ are not significantly different.

A central motivation of this work is that staleness in momentum arises from the change in gradients and targets. In theory, this is especially problematic for function approximators which tend to have interference, such as DNNs (Fort et al., 2019; Bengio et al., 2020), i.e. where taking a gradient step using a single input affects the output for virtually every other input. More interference means that as we change $\theta$, the accumulated gradients computed from past $\theta$s become stale faster. We test this hypothesis by (1) measuring the *drift* of the value function in different scenarios, and (2) computing the cosine similarity between the corrected gradients of (13), $\hat{g}_i^t$, and their true value $g_i^t = \nabla J_i(\theta_t)$.

In Figure 5 we compare the value drift of MLPs with that of linear models with Radial Basis Function (RBF) features. We compute the value drift of the target on recently seen examples, i.e. we compute the average $(V_{\theta_i}(s_i') - V_{\theta_t}(s_i'))^2$ for the last $h = 2n_{mb}/(1-\beta)$ examples, where $n_{mb}$ is the minibatch size. We compute the RBF features for $s \in \mathbb{R}^2$ as $\exp(-\|s - u_{ij}\|^2/\sigma^2)$ for a regular grid of $u_{ij} \in \mathbb{R}^2$ in the input domain. We find that the methods we try (even the oracle) are all virtually identical when using RBFs (see Figure 12 in appendix). We also find, as shown in Figure 5, that RBFs have very little value drift (due to their sparsity) compared to MLPs. This is consistent with our hypothesis that the method we propose is only useful if there is value drift–otherwise, there is no **optimization bias** incurred by using momentum. We can test this hypothesis further by artificially increasing the width of the RBFs, $\sigma^2$, such that they overlap. As predicted, we find that reducing sparsity (increasing interference) increases value drift and increases the gap between our method and the baseline (Figure 6). This drift correlates with performance when changing $\sigma^2$ (Figure 7).

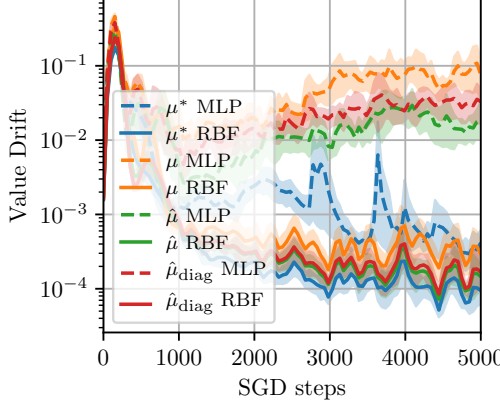

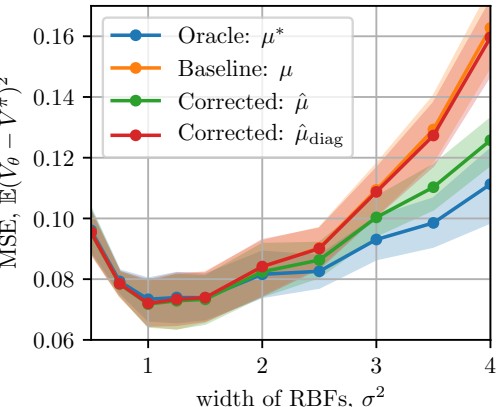

Figure 5: **Value drift** of $V(s')$ when training with TD(0) on a replay buffer. We see that RBFs being a sparse feature representation, the value functions of recently seen data tend not to drift (10 seeds per setting). Here $\sigma^2 = 1$.

Figure 6: MSE as a function of the **width**, $\sigma^2$, of RBF kernels. The larger the kernel, the more value drift our method, $\hat{\mu}$, is able to correct (10 seeds per setting).

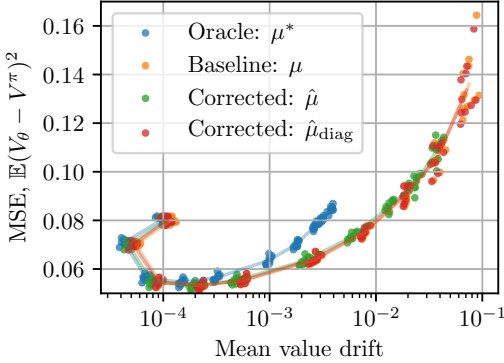

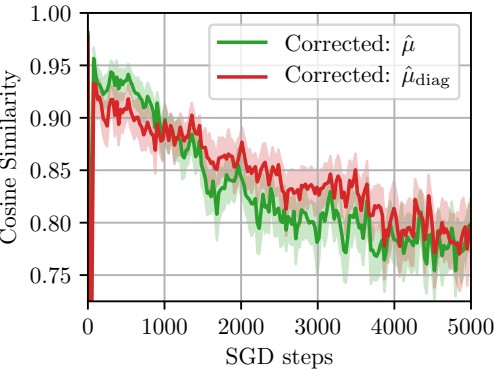

Figure 7: MSE as a function of mean **Value drift** of $V(s')$ for RBFs of varying kernel size. The lines match the $\sigma^2$ lines of Figure 6, and show the relation between $\sigma^2$, drift, and error.

Figure 8: Average **cosine similarity** of the Taylor approximations $\hat{g}_i^t$ with their true value $g_i^t$ for recently seen data; Mountain Car, replay buffer policy evaluation (40 seeds per setting).

In Figure 8 we measure the cosine similarity of the Taylor approximations $\hat{g}_i^t = g_i + Z_i^\top(\theta_t - \theta_i)$ with the true gradients $g_i^t = \nabla J_i(\theta_t)$ for the last $h = 2n_{mb}/(1-\beta)$ examples. We find that the similarity is relatively high (close to 1) throughout training but that it gets lower as the model converges to the true value $V^\pi$. This is also consistent with our hypothesis that there is change (staleness) in gradients, while also validating the approach of using Taylor approximations to achieve this correction mechanism.

It is also possible to correct momentum for a subset of parameters; we try this on a per-layer basis and find that, perhaps counter-intuitively[3], it is *much* better to correct the bottom layers (close to $x$) than the top layers (close to $V$), although correcting all layers is the best (see appendix Figure 13).

We now apply our method on the Atari MsPacman game. We first do **policy evaluation** on an expert agent (we use a pretrained Rainbow agent (Hessel et al., 2018)). Since the architecture required to train an Atari agent is too large ($n \approx 4.8M$) to maintain $n^2$ values, we only use the diagonal version of our method. We also experiment with smaller ($n \approx 12.8k$) models and the full correction,

---

[3]Although one may expect that changes close to $V$ should produce more drift (it is commonly thought that bootstrapping happens in the last layer on top of relatively stable representations), the opposite is consistent with $\hat{\mu}$ interacting with *interference* in the input space, which the bottom layers have to learn to disentangle.

with similar results (see D.8). To (considerably) speed up learning, since the model is large, we additionally use per-parameter learning rates as in Adam (Kingma & Ba, 2015), where an estimate of the second moment is used as denominator in the parameter update; we denote this combination $\hat{\mu}_{\text{diag}}/\sqrt{\nabla^2 + \epsilon}$. We see in Figure 9 that our method provides a significant ($p < 0.01$) advantage.

Note that our method does not use frozen targets (as is usually necessary for this environment). A simple way to avoid momentum staleness and/or target drift in TD(0) is the use of **frozen targets**, i.e. to keep a separate $\bar{\theta}$ to compute $V_{\bar{\theta}}(s')$, updated ($\bar{\theta} \leftarrow \theta$) at large intervals. Such a method is central to DQN (Mnih et al., 2013), but its downside is that it requires more updates to bootstrap. We find that for *policy evaluation*, frozen targets are much slower (both in Atari and simple environments) than our baseline (see appendix Figures 16 and 17).

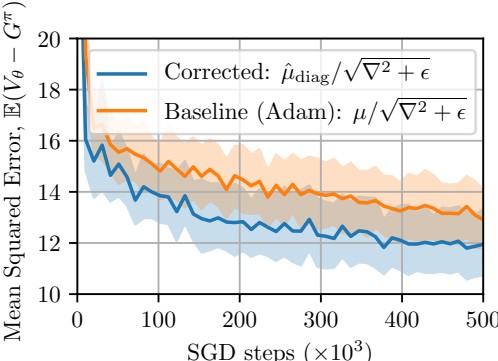

Figure 9: TD(0) policy evaluation on Atari (MsPacman) with varying momentums (20 seeds) on a **replay buffer**. The MSE is measured against sampled returns $G^\pi$.

We finally apply our method to **control**, first in combination with a **policy gradient** method, PPO (Schulman et al., 2017), in its policy evaluation part, and second with a 5-step **Sarsa** method, but find no improvement (or marginal at best) in either setting. As in simpler environments we measure cosine similarity and value drift. We find low similarity ($\approx 0.1$) but a $2\times$ to $3\times$ decrease in drift using our method, suggesting that while our method corrects drift, its effect on policy improvement is minor. We suspect that in control, even with a better value function, other factors such as exploration or overestimation come into play which are not addressed by our method.

## 4 DISCUSSION

We propose a method which improves momentum, when applied to DNNs doing TD learning, by correcting gradients for their staleness via an approximate Taylor expansion. We show that correcting this staleness is particularly useful when learning online using a TD objective, but less so for supervised learning tasks. We show that the proposed method corrects for value drift in the bootstrapping target of TD, and that the proposed approximate Taylor expansion is a useful tool that aligns well enough with the true gradients.

**Shortcomings of the method** In its most principled form, the proposed method requires computing a second-order derivative, which is impractical in most deep learning settings. While we do find experimentally that ignoring its contribution has a negligible effect in toy settings, we are unable to verify this for larger neural networks. Compared to the usual momentum update, the proposed method also requires performing two additional backward passes, as well as storing $2n + n^2$ additional scalars. This can be improved with a diagonal approximation requiring only $3n$ scalars, but this approximation is not as precise, which is currently an obstacle for large architectures. While these extra computations improve *sample complexity*, i.e. we get more out of each sample, they do not necessarily improve convergence speed in *computation time* (although, our method is suited to GPU parallelism and has reasonable speed even for large $n^2$s, including in Atari). Sample complexity may be particularly important in settings where samples are expensive to get (e.g. real world systems), but less so when they are almost free (e.g. simulations).

**Incorporating TD in optimization** One meta-result stands out from this work: something is lost when naively applying supervised learning optimization tools to RL algorithms. In particular here, by simply taking into account the non-stationarity of the TD objective, we successfully improve the sample complexity of a standard tool (momentum), and demonstrate the potential value of incorporating elements of RL into supervised learning tools. The difference explored here with momentum is only one of the many differences between RL and supervised learning, and we believe there remain plenty of opportunities to improve deep learning methods by understanding their interaction with the peculiarities of RL.

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

# A    RELATED WORK

To the best of our knowledge, no prior work attempts to derive a corrected momentum-SGD update adapted to the Temporal Difference method. That being said, a wealth of papers are looking to accelerate TD and related methods.

**On momentum, traces, and gradient acceleration in TD**

From an RL perspective, our work has some similarity to the so-called eligibility traces mechanism. In particular, in the True Online TD($\lambda$) method of van Seijen & Sutton (2014), the authors derive a *strict-online* update (i.e. weights are updated at every MDP step, using only information from past steps, rather than future information as in the $\lambda$-return perspective) where the main mechanism of the derivation lies in finding an update by assuming (at least analytically) that one can "start over" and reuse all past data iteratively at each step of training, and then from this analytical assumption derive a recursive update (that doesn't require iterating through all past data). The extra values that have to be kept to compute the recursive updates are then called traces. This is akin to how we conceptualize $\mu^*$, (6), and derive $\hat{\mu}$.

The conceptual similarities of the work of van Seijen & Sutton (2015) with our work are also interesting. There, the authors analyse what "retraining from scratch" means (i.e., again, iteratively restarting from $\theta_0 \in \mathbb{R}^m$) but with some ideal target $\theta^*$ (e.g. the current parameters) by redoing sequentially all the TD(0) updates using $\theta^*$ for all the $n$ transitions in a replay buffer, costing $O(nm)$. They derive an online update showing that one can continually learn at a cost of $O(m^2)$ rather than paying $O(nm)$ at each step. The proposed update is also reminiscent of our method in that it aims to perform an approximate batch update without computing the entire batch gradient, and also maintains extra momentum-like vectors and matrices. We note that the derivation there only works in the linear TD case.

In a way, such an insight can be found in the original presentation of TD($\lambda$) of Sutton (1988), where the TD($\lambda$) parameter update is written as (equation (4) in the original paper, but with adapted notation):

$$\Delta\theta_t = \alpha[r_t + \gamma V_{\theta_t}(s_{t+1}) - V_{\theta_t}(s_t)] \sum_{k=1}^{t} \lambda^{t-k} \nabla_{\theta_t} V_{\theta_t}(s_k)$$

Remark the use of $\theta_t$ in the sum; in the linear case since $\nabla_{\theta_t} V_{\theta_t}(s_k) = \phi(s_k)$, the sum does not depend on $\theta_t$ and thus can be computed recursively. A posteriori, if one can find a way to cheaply compute $\nabla_{\theta_t} V_{\theta_t}(s_k) \ \forall k$, perhaps using the method we propose, it may be an interesting way to perform TD($\lambda$) using a non-linear function approximator.

Our analysis is also conceptually related to the work of Schapire & Warmuth (1996), where a worst-case analysis of TD*($\lambda$) is performed using a *best-case learner* as the performance upper bound. This is similar to our *momentum oracle*; just as the momentum oracle is the "optimal" approximation of the accumulation gradients coming from all past training examples, the best-case learner of Schapire & Warmuth (1996) is the set parameters that is optimal when one is allowed to look at all past training examples (in contrast to an online TD learner).

Before moving on from TD($\lambda$), let us remark that eligibility traces and momentum, while similar, estimate different quantities. The usual (non-replacing) traces estimate the exponential moving average of the gradient of $V_\theta$, while momentum does so for the objective $J$ (itself a function of $V_\theta$):

$$\mathbf{e}_t = (1 - \lambda) \sum_{k}^{t} \lambda^{t-k} \nabla_\theta V_\theta, \quad \mu_t = (1 - \beta) \sum_{k}^{t} \beta^{t-k} \nabla_\theta J(V_\theta)$$

Our method also has similarities with residual gradient methods (Baird, 1995). A recent example of this is the work of Zhang et al. (2019), who adapt the residual gradient for deep neural networks. Residual methods learn by taking the gradient of the TD loss with respect to both the current value and the next state value $V(S')$, but this comes at the cost of requiring two independent samples of $S'$ (except in deterministic environments).

Similarly, our work is related to the "Gradient TD" family of methods (Sutton et al., 2008; 2009). These methods attempt to maintain an expectation (over states) of the TD update, which allows to

directly optimize the Bellman objective. While the exact relationship between GTD and "momentum TD" is not known, they both attempt to maintain an "expected update" and adjust parameters according to it; the first approximates the one-step linear TD solution, while the latter approximates the one-step batch TD update. Note that linear GTD methods can also be accelerated with momentum-style updates (Meyer et al., 2014), low-rank approximations for part of the Hessian (Pan et al., 2016), and adaptive learning rates (Gupta et al., 2019).

More directly related to this work is that of Sun et al. (2020), who show convergence properties of a rescaled momentum for linear TD(0). While most (if not every) deep reinforcement learning method implicitly uses some form of momentum and/or adaptive learning rate as part of the deep learning toolkit, Sun et al. (2020) properly analyse the use of momentum in a (linear) TD context. Gupta (2020) also analyses momentum in the context of a linear TD(0) and TD($\lambda$), with surprising negative results suggesting naively applying momentum may hurt stability and convergence in minimal MDPs.

Another TD-aware adaptive method is that of Romoff et al. (2020), who derive per-parameter adaptive learning rates, reminiscent of RMSProp (Hinton et al., 2012), by considering a (diagonal) Jacobi preconditioning that takes into account the bootstrap term in TD.

Finally, we note that, as far as we know, recent deep RL works all use some form of adaptive gradient method, Adam (Kingma & Ba, 2015) being an optimizer of choice, closely followed by RMSProp (Hinton et al., 2012); notable examples of such works include those of Mnih et al. (2013), Schulman et al. (2017), Hessel et al. (2018), and Kapturowski et al. (2019). We also note the work of Sarigül & Avci (2018), comparing various SGD variants on the game of Othello, showing significant differences based on the choice of optimizer.

**On Taylor approximations** Balduzzi et al. (2017) note that while theory suggests that Taylor expansions around parameters should not be useful because of the "non-convexity" of ReLU neural networks, there nonetheless exists local regions in parameter space where the Taylor expansion is consistent. Much earlier work by Engelbrecht (2000) also suggests that Taylor expansions of small sigmoid neural networks are easier to optimize. Using Taylor approximations around parameters to find how to prune neural networks also appears to be an effective approach with a long history (LeCun et al., 1990; Hassibi & Stork, 1993; Engelbrecht, 2001; Molchanov et al., 2016).

**On policy-gradient methods and others** While not discussed in this paper, another class of methods used to solve RL problems are PG methods. They consist in taking gradients of the objective wrt a directly parameterized policy (rather than inducing policies from value functions). We note in particular the work of Baxter & Bartlett (2001), who analyse the bias of momentum-like cumulated policy gradients (referred to as traces therein), showing that $\beta$ the momentum parameter should be chosen such that $1/(1 - \beta)$ exceeds the mixing time of the MDP.

Let us also note the method of Vieillard et al. (2020), Momentum Value Iteration, which uses the concept of an exponential moving average objective for a decoupled (with its own parameters) action-value function from which the greedy policy being evaluated is induced. This moving average is therein referred to as *momentum*; even though it is not properly speaking the optimizational acceleration of Polyak (1964), its form is similar.

## B COMPLETE DERIVATIONS

### B.1 DERIVATION OF MOMENTUM CORRECTION

In momentum, $\mu_t$ can be rewritten as:

$$\mu_t = (1 - \beta) \sum_{i=1}^{t} \beta^{t-i} \nabla_{\theta_i} J_i(\theta_i) \tag{21}$$

We argue that an ideal *unbiased* momentum $\mu_t^*$ would approximate the large batch case by only discounting past minibatches and using current parameters $\theta_t$ rather than past parameters $\theta_i$:

$$\mu_t^* \overset{\text{def}}{=} (1 - \beta) \sum_{i=1}^{t} \beta^{t-i} \nabla_{\theta_t} J_i(\theta_t) \tag{22}$$

Note that the only difference between (21) and (22) is the use of $\theta_i$ vs $\theta_t$. The only way to compute $\mu_t^*$ exactly is to recompute the entire sum after every parameter update. Alternatively, we could somehow approximate this sum. Below we will define $g_i^t = \nabla_{\theta_t} J_i(\theta_t)$, which we will then approximate with $\hat{g}_i^t$. We will then show that this approximation has a recursive form which leads to an algorithm.

We want to correct accumulated past gradients $g_i$ to their "ideal" form $g_i^t$, as above. We do so with their Taylor expansion around $\theta$, and we write the correction of the gradient $g_i$ computed at learning time $i$ corrected at time $t$ as:

$$g_i^t = g_i(\theta_i + \Delta\theta(t-1;i)) = g_i + \nabla_\theta g_i^T \Delta\theta(t-1;i) + o(\|\Delta\theta\|_2^2) \tag{23}$$

$$\approx \hat{g}_i^t = g_i + Z_i^T \Delta\theta \tag{24}$$

where $\Delta\theta(t;i) = \theta_t - \theta_i$, $g_i = \nabla_{\theta_i} J_i$, $Z_i = \nabla_{\theta_i} g_i$.

Here we are agnostic of the particular form of $Z$, which will depend on the loss and learning algorithm, and is *not necessarily* the so-called Hessian. To see why this is the case, and for the derivation of $Z$ for the squared loss, cross-entropy and TD(0), see section B.2.

Let's now express $\hat{g}_i^t$ in terms of $g_i$ and updates $\mu_t$. At each learning step, the current momentum $\mu_t$ is multiplied with the learning rate $\alpha$ to update the parameters, which allows us to more precisely write $\hat{g}_i^t$:

$$\theta_t = \theta_{t-1} - \alpha\mu_t = \theta_0 - \alpha\sum_{i=1}^{t}\mu_i \tag{25}$$

$$\Delta\theta(t;i) = \theta_t - \theta_i = \theta_0 - \alpha\sum_{k=1}^{t}\mu_k - \theta_0 + \alpha\sum_{k=1}^{i}\mu_k = -\alpha\sum_{k=i}^{t}\mu_k \tag{26}$$

$$\hat{g}_i^t = g_i + Z_i^T\Delta\theta(t-1;i) = g_i - \alpha Z_i^T\sum_{k=i}^{t-1}\mu_k \tag{27}$$

We can now write $\hat{\mu}_t$, the approximated $\mu_t^*$ using $\hat{g}_i^t$:

$$\hat{\mu}_t = (1-\beta)g_t + (1-\beta)\sum_{i=1}^{t-1}\beta^{t-i}\hat{g}_i^t \tag{28}$$

$$= (1-\beta)g_t + (1-\beta)\sum_{i=1}^{t-1}\beta^{t-i}g_i - (1-\beta)\sum_{i=1}^{t-1}\beta^{t-i}\alpha Z_i^T\sum_{k=i}^{t-1}\hat{\mu}_k \tag{29}$$

$$= \mu_t - \alpha(1-\beta)\sum_{i=1}^{t-1}\sum_{k=i}^{t-1}\beta^{t-i}Z_i^T\hat{\mu}_k \quad \text{extract } \mu \tag{30}$$

$$= \mu_t - \alpha(1-\beta)\sum_{k=1}^{t-1}\sum_{i=1}^{k}\beta^{t-i}Z_i^T\hat{\mu}_k \quad \text{change the sum indices for convenience} \tag{31}$$

$$= \mu_t - \eta_t \quad \text{extract } \eta_t \tag{32}$$

Let's try to find a recursive form for $\eta_t$:

$$
\begin{aligned}
\eta_t - \eta_{t-1} &= \alpha(1-\beta)\sum_{k=1}^{t-1}\sum_{i=1}^{k}\beta^{t-i}Z_i^T\hat{\mu}_k - \alpha(1-\beta)\sum_{k=1}^{t-2}\sum_{i=1}^{k}\beta^{t-1-i}Z_i^T\hat{\mu}_k \\
&= \alpha(1-\beta)\left(\sum_{k=1}^{t-2}\sum_{i=1}^{k}(\beta^{t-i}Z_i^T\hat{\mu}_k - \beta^{t-1-i}Z_i^T\hat{\mu}_k) + \sum_{i=1}^{t-1}\beta^{t-i}Z_i^T\hat{\mu}_{t-1}\right) \\
&= \alpha(1-\beta)\left(\sum_{k=1}^{t-2}\sum_{i=1}^{k}(\beta-1)\beta^{t-1-i}Z_i^T\hat{\mu}_k + \sum_{i=1}^{t-1}\beta^{t-i}Z_i^T\hat{\mu}_{t-1}\right) \\
&= (\beta-1)\eta_{t-1} + \alpha\beta(1-\beta)\sum_{i=1}^{t-1}\beta^{t-1-i}Z_i^T\hat{\mu}_{t-1}
\end{aligned}
$$

$$
\text{Let}\qquad \zeta_t = (1-\beta)\sum_{i=1}^{t}\beta^{t-i}Z_i
$$

$$
\begin{aligned}
&= (\beta-1)\eta_{t-1} + \alpha\beta\zeta_{t-1}\hat{\mu}_{t-1} \\
\eta_t - \eta_{t-1} &= -(1-\beta)\eta_{t-1} + \alpha\beta\zeta_{t-1}^T\hat{\mu}_{t-1} \\
\eta_t &= \beta\eta_{t-1} + \alpha\beta\zeta_{t-1}^T\hat{\mu}_{t-1}
\end{aligned}
$$

We can now write the full update as:

$$
\begin{aligned}
\hat{\mu}_t &= \mu_t - \eta_t \\
\eta_t &= \beta\eta_{t-1} + \alpha\beta\zeta_{t-1}^T\hat{\mu}_{t-1} \\
\mu_t &= (1-\beta)\sum_{i=1}^{t}\beta^{t-i}g_i = (1-\beta)g_t + \beta\mu_{t-1} \\
\zeta_t &= (1-\beta)\sum_{i=1}^{t}\beta^{t-i}Z_i = (1-\beta)Z_t + \beta\zeta_{t-1}
\end{aligned}
$$

with $\eta_0 = \mathbf{0}$.

This corresponds to an algorithm where one maintains $\eta$, $\mu$ and $\zeta$.

## B.2 DERIVATION OF TAYLOR EXPANSIONS AND $Z$

For least squares regression, the expansion around $\theta$ of $g(\theta)$ is simply the Hessian of the loss $\delta^2$:

$$
\delta^2 = \frac{1}{2}(y - f_\theta(x))^2 \tag{33}
$$

$$
g(\theta) = \nabla_\theta\delta^2 \tag{34}
$$

$$
g(\theta + \Delta\theta) = g(\theta) + H_J^T\Delta\theta + o(\|\Delta\theta\|_2^2) \tag{35}
$$

This Hessian has the form

$$
H_J = \nabla_\theta f \otimes \nabla_\theta f + \delta^2\nabla_\theta^2 f \tag{36}
$$

$\delta^2$ being small when a neural network is trained and $\nabla^2$ being expensive to compute, a common approximation to $H_j$ is to only rely on the outer product. Thus we can write:

$$
Z_{reg} = \nabla_\theta f \otimes \nabla_\theta f \tag{37}
$$

For classification, or categorical crossentropy, $Z$ has exactly the same form, but where $f$ is the log-likelihood (i.e. output of a log-softmax) of the correct class.

For TD(0), the expansion is more subtle. Since TD is a semi-gradient method, when computing $g(\theta)$, gradient of the TD loss $\delta^2$, we ignore the bootstrap target's derivative, i.e. we hold $V_\theta(s')$ constant (unlike in GTD). On the other hand, when computing the Taylor expansion around $\theta$, we do care about how $V_\theta(s')$ changes, and so its gradient comes into play:

$$
\begin{aligned}
g(\theta) &= (V_\theta(x) - \gamma V_\theta(x') - r)\nabla_\theta V_\theta(x) \\
g(\theta + \Delta\theta)_i &= g_i(\theta) + (\nabla_\theta V_\theta(x) - \gamma\nabla_\theta V_\theta(x'))\nabla_{\theta_i} V_\theta(x) \cdot \Delta\theta + \delta\nabla_\theta\nabla_{\theta_i} V_\theta(x) \cdot \Delta\theta \\
g(\theta + \Delta\theta) &= g(\theta) + ((\nabla_\theta V_\theta(x) - \gamma\nabla_\theta V_\theta(x')) \otimes \nabla_\theta V_\theta(x))^T \Delta\theta + \delta\nabla_\theta^2 V_\theta(x)^T \Delta\theta
\end{aligned}
$$

Similarly for TD we ignore the second order derivatives and write:

$$
Z_{TD} = (\nabla_\theta V_\theta(x) - \gamma\nabla_\theta V_\theta(x')) \otimes \nabla_\theta V_\theta(x)
$$

For an $n$-step TD objective, the correction is very similar:

$$
Z_{TD(n)} = (\nabla_\theta V_\theta(x_t) - \gamma^n\nabla_\theta V_\theta(x_{t+n})) \otimes \nabla_\theta V_\theta(x_t)
$$

For a forward-view TD($\lambda$) objective this is also similar, but more expensive:

$$
Z_{TD(\lambda)} = (\nabla_\theta V_\theta(x_t) - (1-\lambda)\nabla_\theta(\gamma\lambda V_\theta(x_{t+1}) + \gamma^2\lambda^2 V_\theta(x_{t+2}) + ...)) \otimes \nabla_\theta V_\theta(x_t)
$$

Finally, to avoid maintaining $n \times n$ values for $Z$, it is possible to only maintain the diagonal of $Z$ or some block-diagonal approximation, at some performance cost.

## B.3 THE LINEAR CASE

Much of RL analysis is done in the linear case, as it is easier to make precise statements about convergence there. We write down such a case below, but we were unfortunately unable to derive any interesting analyses from it.

Recall that the proposed method attempts to approximate $\mu^*$:

$$
\mu_t^* = (1-\beta)\sum_{i=1}^t \beta^{t-i}\nabla_{\theta_t} J_i(\theta_t)
$$

which in the linear case $V_\theta(x) = \theta^\top\phi(x)$ is simply:

$$
= (1-\beta)\sum_{i=1}^t \beta^{t-i}\delta_i\phi_i
$$

which depends on $\theta_i$ through $\delta_i$ the TD error. As such we can write $Z$ as:

$$
Z_{TD} = (\nabla_\theta V_\theta(x) - \gamma\nabla_\theta V_\theta(x')) \otimes \nabla_\theta V_\theta(x) = (\phi - \gamma\phi')\phi^\top
$$

which interestingly does not depend on $\theta$. To the best of our knowledge, and linear algebra skills, this lack of dependence on $\theta$ does *not* allow for a simplification of the proposed correction mechanism (in contrast with linear eligibility traces) due to the pairwise multiplicative $t, i$ dependencies that emerge.

## B.4 MINIBATCHING

It is possible to only do 3 (or 2 in the supervised case) "backward" calls (e.g. `torch.autograd.grad` in PyTorch) even when computing this correction for several examples. This is in contrast to other adaptive gradient methods (Romoff et al., 2020) which require per-input

gradients (these can be computed efficiently with methods such as Dangel et al. (2020), but remain more expensive).

Because the gradient we care to correct is the gradient of the minibatch, i.e. the sum of gradients, then we only have to correct that sum, and not individual gradients. This allows us to compute $Z$ out of the gradients of the minibatch instead of the individual gradients. When the gradient is:

$$g_t = \nabla_\theta J_t(\theta) = \nabla_\theta \sum_i^m J_{t,i}(\theta) = \sum_i^m \nabla_\theta J_{t,i}(\theta) \tag{38}$$

then the first derivative used in its Taylor expansion is, by linearity of the derivative:

$$\frac{\partial g_t}{\partial \theta} = \sum_i^m \nabla_\theta f_i \otimes \nabla_\theta f_i + \delta_{t,i} H_f \tag{39}$$

$$= \nabla_\theta(\sum f_i) \otimes \nabla_\theta(\sum f_i) + \nabla_\theta^2 J_{t,i} \tag{40}$$

Ignoring the Hessian we only have to use the gradients obtained for the sum (or the mean if $J$ is a mean) of losses of the minibatch, which requires only one backpropagation regardless of batch size. The derivation is analogous for TD(0). Note that this would not work if we used the second order derivative due to the multiplicative $\delta_{t,i}$ factors.

## C  ADDITIONAL FIGURES

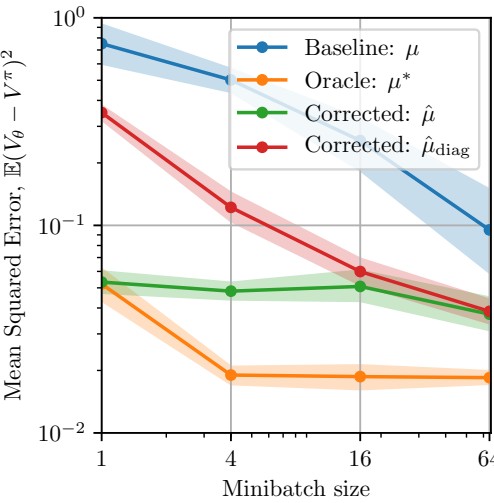

Figure 10: TD(0) policy evaluation on Mountain Car with varying minibatch size on a **replay buffer**. The MSE is measured after 5k SGD steps against a pretrained $V^\pi$. Shaded areas are bootstrapped 95% confidence runs (20 seeds per setting).

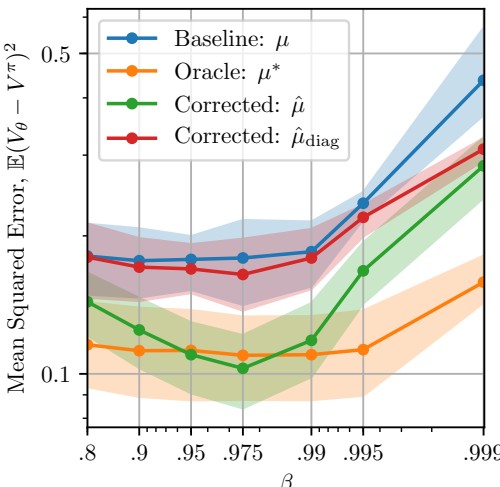

Figure 11: TD(0) policy evaluation on Mountain Car with varying $\beta$ on a **replay buffer**. The MSE is measured after 5k SGD steps against a pretrained $V^\pi$. Shaded areas are bootstrapped 95% confidence runs (10 seeds per setting). We use a minibatch size of 4 to reveal interesting trends.

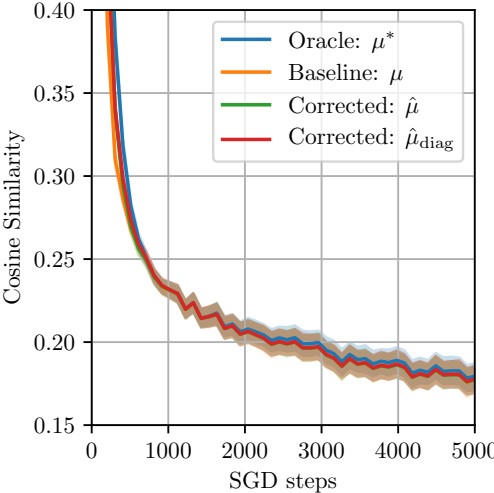
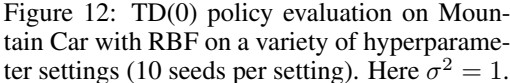
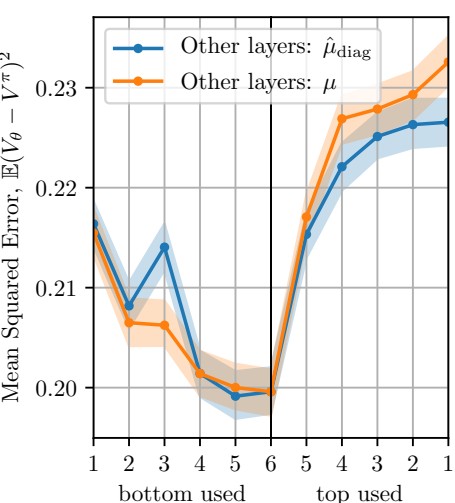

Figure 12: TD(0) policy evaluation on Mountain Car with RBF on a variety of hyperparameter settings (10 seeds per setting). Here $\sigma^2 = 1$.

Figure 13: TD(0) policy evaluation on Mountain Car with an MLP. We vary the number of layers whose parameters are used for full $\hat{\mu}$ correction ($n^2$ params); e.g. when "bottom used" is 3, the first 3 layers, those closest to the input, are used; when "top used" is 1, only the last layer, that predicts $V$ from embeddings, is used. The parameters of other layers are either corrected with the diagonal correction or use normal momentum. Correcting "both ends" is not better than just the bottom (not shown here).

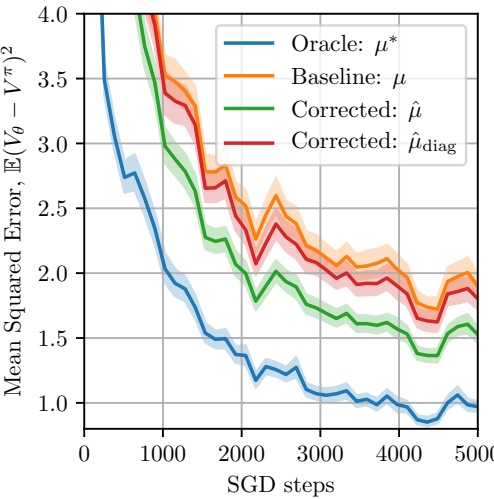
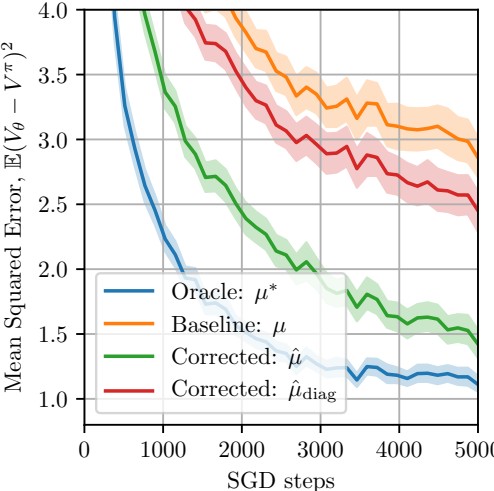

Figure 14: TD(0) policy evaluation on Acrobot with varying hyperparameters on a **replay buffer**. The MSE is measured after 5k SGD steps against a pretrained $V^\pi$. Shaded areas are bootstrapped 95% confidence runs (10 seeds per setting).

Figure 15: TD(0) policy evaluation on Cartpole with varying hyperparameters **replay buffer**. The MSE is measured after 5k SGD steps against a pretrained $V^\pi$. Shaded areas are bootstrapped 95% confidence runs (10 seeds per setting).

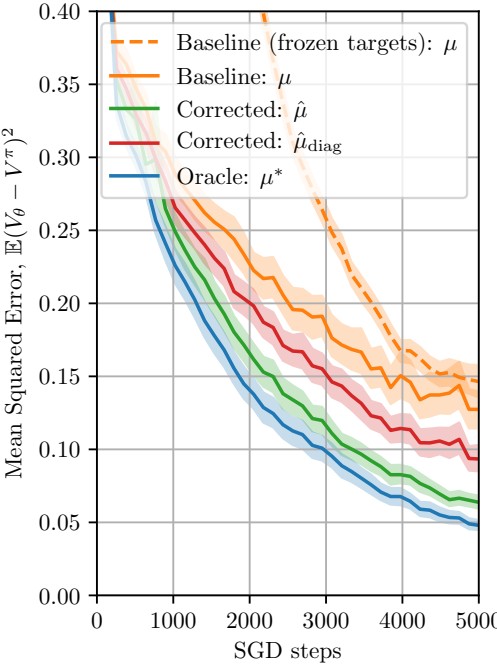

Figure 16: Replication of Figure 3 including the frozen targets baseline.

Figure 17: Replication of Figure 9 including the frozen targets baseline. Interestingly the models trained with frozen targets eventually become more precise than those without, but this only happens after a very long time. This is explained by the stability required for bootstrapping when TD errors become increasingly small, which is easily addressed by keeping the target network fixed.

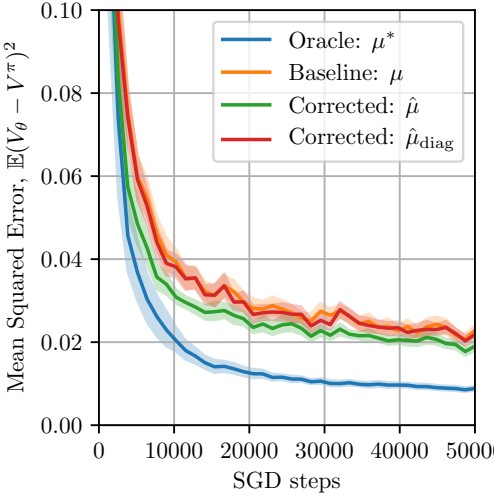

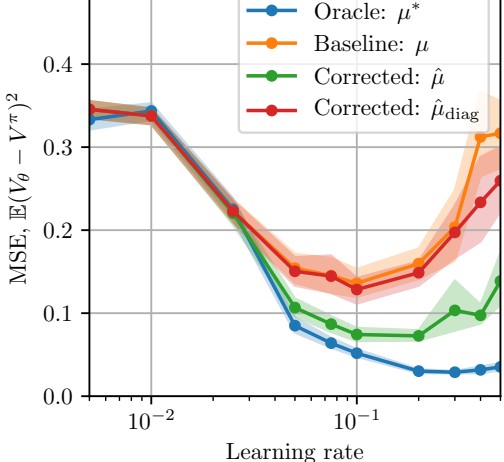

Figure 18: Replication of Figure 3 with $10\times$ more training steps. Methods gradually converge to the value function.

Figure 19: Effect of the learning rate on Mountain Car, replay buffer policy evaluation, MSE after 5k training steps.

## D    HYPERPARAMETER SETTINGS AND COMMENTS

All experiments are implemented using PyTorch (Paszke et al., 2019). We use Leaky ReLUs throughout. All experimental code is available in supplementary materials.

On Leaky ReLUs: we did experiment with ReLU, tanh, ELU, and SELU activation units. The latter 3 units have more stable Taylor expansions for randomly initialized neural networks, but in terms of experimental results, Leaky ReLUs were always significantly better.

### D.1    FIGURE 1

We use an MLP with 4 layers of width $n_h$.

We use the cross-product of $n_h \in \{8, 16, 32\}$, $\beta \in \{0.9, 0.99\}$, $\alpha \in \{0.005, 0.01\}$, $n_{mb} \in \{4, 16, 64\}$.

### D.2    FIGURE 2

We use a convolutional model with the following sequence of layers, following PyTorch convention: Conv2d(3, $n_h$, 3, 2, 1), Conv2d($n_h$, $2n_h$, 3, 2, 1), Conv2d($2n_h$, $2n_h$, 3, 2, 1), Conv2d($2n_h$, $n_h$, 3, 1, 1), Flatten(), Linear($16n_h$, $4n_h$ ), Linear($4n_h$, $4n_h$), Linear($4n_h$, 10), with LeakyReLUs between each layer.

We use the cross-product of $n_h \in \{8, 16\}$, $\beta \in \{0.9, 0.99\}$, $\alpha \in \{0.005, 0.01\}$, $n_{mb} \in \{4, 16, 64\}$.

### D.3    FIGURE 3 AND 16

We use an MLP with 4 layers of width $n_h$.

We use the cross-product of $n_h \in \{8, 16, 32\}$, $\beta \in \{0.9, 0.99\}$, $\alpha \in \{0.5, 0.1, 0.05\}$, $n_{mb} \in \{4, 16, 64\}$.

### D.4    FIGURE 4

We use an MLP with 4 layers of width $n_h$.

We use the cross-product of $n_h \in \{16\}$, $\beta \in \{0.9, 0.99\}$, $\alpha \in \{0.005, 0.001, 0.0005\}$, $n_{mb} \in \{1\}$.

### D.5    FIGURE 5 AND 12

We use a linear layer on top of a grid RBF representation with each gaussian having a variance of $\sigma^2/n_{grid}$.

For the RBF we use $n_{grid} = 20$, $\alpha = 0.1$, $n_{mb} = 16$, $\beta = 0.99$. For the MLP we use $n_h = 16$, $\alpha = 0.1$, $n_{mb} = 16$, $\beta = 0.99$.

### D.6    FIGURE 6 AND 7

We use RBFs with $\sigma^2 \in \{4, 3.5, 3, 2.5, 2, 1.5, 1.25, 1, 0.75, 0.5\}$, $n_{grid} \in \{10, 20\}$, $\alpha \in \{0.1, 0.01\}$.

### D.7    FIGURE 8

We use an MLP with 4 layers of width $n_h = 16$, $\alpha = 0.1$, $n_{mb} = 16$, $\beta = 0.95$.

### D.8    FIGURE 9 AND 17

We use the convolutional model of Mnih et al. (2013) with the same default hyperparameters, following PyTorch convention: Conv2d(4, $n_h$, 8, stride=4, padding=4), Conv2d($n_h$, $2n_h$, 4, stride=2, padding=2), Conv2d($2n_h$, $2n_h$, 3, padding=1), Flatten(), Linear($2n_h \times 12 \times 12$, $16n_h$), Linear($16n_h$, $n_{acts}$).

We use $n_h = 64$, $n_{mb} = 32$, for Adam we use $\alpha = 5 \times 10^{-5}$ and $\beta = 0.99$, for our method we use $\alpha = 10^{-4}$ and $\beta = 0.9$ (these choices are the result of a minor hyperparameter search of which the best values were picked, equal amounts of compute went towards our method and the baseline so as to avoid "poor baseline cherry picking"). For the frozen target baseline we update the target every 2.5k steps.

We were able to find similar differences between Adam and our method with a much smaller model, but using the full correction instead of the diagonal one. Although the full correction outperforms Adam when both use this small model, using so few parameters is not as accurate as the original model described above, and we omit these results.

This small model is: Conv2d(4, $n_h$, 3, 2, 1), Conv2d($n_h$, $n_h$, 3, 2, 1), Conv2d($n_h$, $n_h$, 3, 2, 1), Conv2d($n_h$, $n_h$, 3, 2, 1), Conv2d($n_h$, $n_{acts}$, 6). For $n_h = 16$ (which we used for experiments), this model has a little less than 16k parameters, making $Z$ about 250M scalars. While this is large, this easily fits on a modern GPU, and the extra computation time required for the full correction mostly still comes from computing two extra backward passes, rather than from computing $Z$ and the correction.

This is beyond the scope of this paper, but is worth of note: Interestingly this small model still works quite well for control (both with Adam and our method). We have not tested this extensively, but, perhaps contrary to popular RL wisdom surrounding Deep Q-Learning, we were able to train decent MsPacman agents with (1) no replay buffer, but rather 4 or more parallel environments (the more the better) as in A2C (Mnih et al., 2016) (2) no frozen target network (3) a Q network with only 16k parameters rather than the commonplace 4.8M-parameter Atari DQN model. The only "trick" required is to use 5-step Sarsa instead of 1-step TD (as suggested by the results of Fedus et al. (2020), although in their case a replay buffer is used).

### D.9 FIGURE 10

We use the same configuration than Figure 3 with $n_{mb} \in \{1, 4, 16, 64\}$.

### D.10 FIGURE 11

We use the same configuration than Figure 3 with $\beta \in \{.8, .9, .95, .975, .99, .995, .999\}$.

### D.11 FIGURE 13

We use an MLP with 6 layers of width $n_h = 16$, $\alpha = 0.1$, $n_{mb} = 16$, $\beta = 0.9$. We vary which layers get used in the corrected momentum; see caption.

### D.12 FIGURE 14 AND 15

We use the same settings as Figure 3, with the exception that we do 5-step TD in Figure 14, for Acrobot, and 3-step TD for Figure 15, for Cartpole. Using $n > 1$ appears necessary for convergence for both our method and the baseline.

