# OpenReview forum: "Correcting Momentum in Temporal Difference Learning"
_ICLR.cc/2021/Conference — Reject_

### Official Review · AnonReviewer4 · 2020-10-20
**the motivation is not very convincing**

**Rating:** 4
**Confidence:** 4

**Review:**

The paper studies the approach to 'correct the bias' of momentum in supervised learning and reinforcement learning (especially, in the temporal-difference algorithm). The proposed approach can maintain a moving average of the 'correct' gradient direction over past objective functions via extra updates. And the experiments show that it does speed up the convergence. (ps: I did not find the comparisons over the loss functions at the stationary points).

As we know TD learning uses a semi/partial/biased gradient with respect to the mean square Bellman error. Thus, the effect of applying momentum to training is an open question. I feel very glad to see such a paper studying it and adding a reasonable modification. The approach averages the TD update over different steps via a modified momentum update. Such an approach reduces the variance, which is usually very large in TD, but also uses additional computational power.

I found the motivation not very convincing for me. The paper seems to claim that a good momentum should carry the information of past objective function and calculate their gradients only with respect to the parameter in the current iterate. However, momentum terms do have the reason to use the gradient wrt previous parameters, as it can speed up the convergence in many situations, e.g. ravines. Another concern is that TD is not typically an online learning problem. The objective function in the early phase is not informative to the update in the later phases as the values of the next states are inaccurate. Therefore, I feel the theoretical reason for 'correcting' the 'bias', which is defined by this paper, is not warranted very well.

I suggest the paper to rigorously discuss more on why such 'bias' has to be corrected. Alternatively, the proposed method is designed simply for the purpose of reducing the variance of the stochastic gradient for the case of minibatch update. But the paper needs to argue how it is better than using a larger batch or compare itself with other variance reduction approaches.

---

> ### Author Response · Authors · 2020-11-12
> **Response**
>
> Thank you for your comments.
>
> It is indeed on us to explain our motivation better. In this case we respectfully believe that you may not have understood the text correctly, so we will work to make this part clearer. We offer a more complete explanation below by pointing out where we believe there are misunderstandings. We hope it is clearer and that you will find that our work is well motivated.
>
> - On loss at stationary points, in classical environments all the methods seem to converge to the same result. We have added a new figure, Figure 18, which shows longer training.
> - “momentum should carry the information of past objective function”, for TD, that is not quite what we claim, we claim that momentum does carry the information of past _states_ (and their gradients) but that as it is, it carries the _wrong_ information about the past objective function, since that objective function has changed (and we propose a correction to this).
> - “momentum terms do have the reason to use the gradient wrt previous parameters, as it can speed up the convergence in many situations” This conjecture is probably true when the loss landscape is constant, as in supervised learning, but in TD the loss landscape changes at every iteration. Since this conjecture has, to the best of our knowledge, not been proved for stochastic gradient momentum supervised learning, it is not clear that there is a good framework to think about it in the case of an evolving loss landscape, nor that it should readily apply to TD.
> - “Another concern is that TD is not typically an online learning problem”. Our proposed method is agnostic of the data distribution, online or not, in fact we show improvements for both the online case and the offline replay-buffer case (Figures 3 and 4).
> - “The objective function in the early phase is not informative to the update in the later phases as the values of the next states are inaccurate.” This is _precisely_ what our method aims to correct. The gradients accumulated in momentum contain information about the “values of next states” which is not informative (since it is inaccurate, since parameters change). Our proposed method corrects those gradients so that they reflect the latest predictions of the “value of next states” and become accurate.
> - “the paper needs to argue how it is better than using a larger batch or compare itself with other variance reduction approaches.” That is a fair point, the effect of our method indeed lessens as the batch size grows (see Figure 10), but using a large batch size is not always possible (e.g. for online learning) or desirable (large batches are associated with overfitting in deep learning literature). Furthermore, using a large batch does not negate the evolution of the loss landscape due to bootstrapping in TD (which we refer to as value drift). This evolution induces staleness when using momentum, and can still be compensated for via our method, since it exists regardless of batch size.

---

### Official Review · AnonReviewer1 · 2020-10-21
**Interesting idea but needs more investigation**

**Rating:** 6
**Confidence:** 4

**Review:**

#### Idea
The paper explains and studies an interesting issue with momentum in TD learning, which is its staleness while doing TD updates which is in contrary to the supervised learning. However, the paper could benefit from more investigations.

#### Comments
- In section 2.1: It is said that to compute the oracle update, one has to recompute the entire sum. This may take a while to compute and it would be good if they have had provided a computation-time or wall-time comparison will prior methods.
- In section 2.1: To handle the problem of long summation, "an effective horizon" is used. However, there's no explanations given behind this particular choice. Is it considered as a geometric distribution and thus this formula? Or if it's not, where does this formula come from?
- In section 3.2: When the proposed method is applied to the Atari games, the problem of scalability shows itself. As mentioned by the authors: "Since the architecture required to train ...", the scalability of this approach is under question.
- In section 3.2: Regarding this paragraph: "Note that we do not use frozen targets ...". The most important question about this work is its scalability, whereas in a prior approaches like frozen targets there is not such an issue while dealing with staleness issue. As in experiments, authors have not used frozen targets, which seems to be misleading when it comes to the experiments and results. It would be useful to see a comparison between these two approaches since they try to solve a similar problem.

#### Minor issues
- In section 3.1: "We task a ..." needs to be changed to "We take a ..."

- In section 2.2: It would be better if you could give a more in depth explanation for eq 11: “Here the term multiplying ∆θ is not exactly the Hessian…”

---

> ### Author Response · Authors · 2020-11-12
> **Response**
>
> Thank you for your comments.
>
> The two concerns we understand you have is that this method does not scale and that the paper needs more investigations. We address the first below, and note that the suggested investigations already appear in the paper, albeit in the appendix.
>
> - Oracle computation: the oracle is included in experiments as a reference point of the “best possible” scenario. The oracle represents what our method would be like if the Taylor approximation was perfect, it is not meant to be compared in wall-time to other methods or prior work, nor do we recommend using the oracle practically. This is purely a scientific tool.
> - This effective horizon formula is commonly used in RL, and indeed does come from the geometric series. Again, since the oracle exists for purely illustrative and comparative reasons, simply choosing a “long enough” horizon is what matters, we could have just as well chosen $h=512$ out of convenience.
> - On scalability: It is fair to point out that the method we propose is computationally expensive, but in writing this paper we also wanted to invite the reader to take a step back, by pointing out that “deep RL methods are not always best served by directly importing techniques from the supervised setting.” By showing the existence of such an inconsistency, we hope to encourage deep RL researchers to consider what (possibly wrong) assumptions they are (re)using from supervised learning through their tools -- perhaps more than we hope deep RL researchers are going to adopt the particular method we propose.
> “The most important question about this work is its scalability”, we would respectfully disagree, progress in machine learning is not only done by proposing new methods, but also by understanding existing methods and their weaknesses. An important question about any scientific work is, what does it teach us? In this work we provide evidence of a previously unreported phenomenon. We do propose _a_ method to combat this phenomenon, but it would vain of us to claim victory over it; we certainly hope that future methods will be able to address this with better tools than Taylor expansions.
> - “authors have not used frozen targets, which seems to be misleading when it comes to the experiments and results. It would be useful to see a comparison between these two approaches since they try to solve a similar problem.” We did run frozen target experiments, please refer to Figure 16 and 17 for such results. They show that the use of frozen targets considerably slows down learning.
> - “Here the term multiplying ∆θ is not exactly the Hessian…”, what the rest of this sentence in the paper explains is this: the Hessian is the second derivative of the loss $\delta^2$, but that this is not what we want so we must be careful. In $\delta^2$, $V_\theta(s’)$ is held constant, but since we update $\theta$, then $V_\theta$ is going to change. Since we care about how the _gradient_ $g$ changes, rather than how $\delta^2$ changes, and that $g$ depends on $V_\theta(s’)$, then when computing the derivative of the $g$ we must take $V_\theta(s’)$ into account rather than holding it constant (which taking the second derivative of $\delta^2$ would). We will attempt to make this more explicit in the paper.

---

> > ### Comment · AnonReviewer1 · 2020-11-16
> > **Response**
> >
> > I appreciate your response and clarification. I do agree with the fact that machine learning does not specifically flourish by proposing new methods, but on the other hand, I believe any study should be done in depth in order to expand our understanding the most. Also, scalability is not a new concern in machine learning, in particular when there are deep networks involved. Thus, seeing efforts on dealing with scalability and applicability is appreciated.
> >
> > Overall, I found your answers somewhat convincing and I would like to increase my score.
> >
> > Thanks.

---

### Official Review · AnonReviewer2 · 2020-10-28
**Interesting direction; limited improvement;**

**Rating:** 6
**Confidence:** 3

**Review:**

This paper proposes a modification to momentum. Extra terms are added to account for the drift in parameters through the gradient updates that contribute to the momentum. The effect of this modification is studied on supervised learning and TD learning and on different representations. The conclusion is that while the original momentum is good enough for supervised learning, this modification matters when there is bootstrapping, especially on representations with high interference.

Overall, the empirical study highlights a direction for improving TD learning and the experiments are diverse and conclusive. Although the proposed approximations to the extra terms show limited improvement, experiments show that if the extra terms can be approximated more accurately, there will be a considerable gain. Computing the extra terms exactly is computationally expensive but it is possible that better approximations will be introduced later on. There are a few weaknesses that I describe below.

1. There is little discussion on why we would want to account for the drift in parameters in momentum. Would this somehow reduce variance or improve the rate of convergence?

2. Candidate step-sizes are too few and some differences could simply be due to the match between the chosen step-size and an algorithm. Consider sweeping over a finer grid.

3. Section 3.1 says "Note that this choice is purely illustrative and that our findings extend to similar simple functions." Is there evidence on other simple functions?

4. A discussion on different optimizers somewhere in the paper or the appendix is needed to see how the proposed updates compare to existing work.

Minor comments:

1. Section 1.1, problem formulation and definition of V and Q: In a standard RL setting reward is a function of state and action.

2. \theta and V_\theta are not defined. It is better to make it explicit that the function is parameterized by \theta and outputs V_\theta which estimates V^\pi.

---
Update: I have the other reviews and rebuttal. I am still leaning towards acceptance, although I do agree with the other reviews that there is little motivation for the idea. While the experiments show improvement, a discussion on convergence rates or variance could show why one should try this idea in the first place, and if \mu_* (rather than regular momentum) is indeed an ideal that the algorithm should approximate.

---

> ### Author Response · Authors · 2020-11-12
> **Response**
>
> Thank you for your comments.
>
> It is fair to point out that the method we propose is computationally expensive, but in writing this paper we also wanted to invite the reader to take a step back, by pointing out that “deep RL methods are not always best served by directly importing techniques from the supervised setting.” By showing the existence of such an inconsistency, we hope to encourage deep RL researchers to consider what (possibly wrong) assumptions they are (re)using from supervised learning through their tools -- perhaps more than we hope deep RL researchers are going to adopt the particular method we propose.
>
> On weaknesses:
> 1. Discussion on drift: this is mostly discussed at the end of section 1.2, as well as throughout the exposition of experimental results, but indeed is a bit lacking. When we point out that momentum staleness (caused by drift) incurs bias, we imply that this bias adds to the biases of TD, which are commonly understood to slow down convergence or even cause divergence. Thus, reducing this bias should improve the rate of convergence. We will make this explicit.
> 2. Learning rates: We have run more experiments and reported the results in a new appendix figure, Figure 19. We note that the differences between algorithms have been consistent even if we select each method’s optimal learning rate.
> 3. In our initial exploratory testing of this method we tried a variety of simple functions for supervised learning which all had the same behaviour, we could include these results explicitly in the appendix. Note also that this lack-of-effect on supervised learning extends to a task such as SVHN (Figure 2), which seems like more valuable evidence.
> 3. Appendix A contains a fairly thorough review of related methods, but perhaps some methods are missing?
> We did experimentally compare (favourably) to RMSProp and deep GTD in the classical environments, but including these results did not seem relevant to the message of the paper (namely that the _current_ use of momentum in Deep RL has issues).

---

### Official Review · AnonReviewer3 · 2020-11-01
**Correcting momentum in Temporal Difference learning**

**Rating:** 6
**Confidence:** 4

**Review:**

Summary:

This paper extends the idea of momentum, commonly used in optimization literature, to Temporal Difference (TD) learning which is a widely used algorithm for policy evaluation in Reinforcement learning literature. The main challenge in this work is to account for the 'optimization bias' introduced by using momentum based updates. The authors propose to do this via a Taylor approximation to the TD update and empirically show the merits of this idea on some toy data sets as well as on an Atari game.

Reason for score:

The idea of using momentum for TD learning seems quite interesting. However, in my opinion, the paper seems to a somewhat incremental contribution, given the work of Sun et. al. 2020, which gives a theoretical analysis of momentum based updates for TD with linear function approximation.

1) The bias correction via Taylor approximation of the gradients seems new to me. The authors claim that the bias doesn't play a significant role in supervised learning tasks. Can the authors provide some more citations to support this besides the 1-d regression task. For TD learning, the authors don't present results for bias corrections with second order terms. How do we know that we can safely ignore these for large networks? On the other hand, incorporating these terms poses a computational challenge.

2) Prior work of Sun et. al. 2020, which proposes a momentum for a linear function approximation, uses projection to bound the iterates. Is the momentum update even stable for deep neural networks for large problems, without some additional tricks? The authors only test toy examples (the Atari example has some combination of Adam and momentum which makes it ).

3) For the comparison on MsPacman, why is a combination with Adam required? The authors claim it is for speeding up learning. In this case, maybe the authors can have the MSE plot for just Adam updates for a better comparison?

4) Many of the plots are truncated at a few thousand steps. What happens after that? It might be helpful to visualize the loss at limit points that different updates converge towards (roughly). Graphs with more steps might be more useful.

5) The authors claim their method improves sample complexity as compared to say the more widely used trick of freezing network. But Figure 17 shows that models with frozen targets eventually become more precise. This might be a reason why this idea may not perform well in the control setting, as Figure 17 suggests that there is probably a residual bias in all momentum based updates (when used with neural networks).

Overall, I am unsure whether the contributions here are strong enough to justify a paper publication.

--------- Update after author response ---------

I thank the authors for their response. I have updated my score. I am slightly leaning toward acceptance though I think the paper might benefit from a revision based on some of the points raised by other reviewers, including comments about motivation for correcting the bias as well as the scalability (the proposed method requires storing n^2 additional values which is expensive for large networks).

---

> ### Author Response · Authors · 2020-11-12
> **Response**
>
> Thank you for your comments.
>
> It is interesting how differently you have interpreted our work than what we thought its message was. Our primary goal was not to “extend” momentum for TD, after all, virtually every deep RL paper already uses an optimizer with momentum, rather, our goal was to argue that these deep RL papers were using momentum _incorrectly_, and that it is possible to correct it by taking into account the nature of TD learning.
>
> As you point out, there are already many momentum-like methods for TD, eligibility traces and related methods (discussed at length in appendix A), but these are restricted to the linear setting, including the work of Sun et al. The novel claim we make is not that of “using momentum for TD learning”, this has been done for a very long time, but rather that in the non-linear (deep) case there are particularly useful corrections to be made that arise from staleness (itself arising from interference, an important phenomenon in deep nets).
>
>
> 1. “Can the authors provide some more citations to support this besides the 1-d regression task”, Figure 2 shows the effect of our proposed method on an SVHN classifier, and shows that even the oracle has little to no effect. Additionally, momentum in the supervised setting is usually viewed as benefiting from this bias (what we call staleness). This bias is actually useful in that it performs some kind of local ensemble averaging which smooths the optimization landscape. This analogy does not hold in TD where said landscape changes as the target changes.
> “How do we know that we can safely ignore these for large networks?”, it is true that we may be missing valuable information, but consider that using the outer product approximation of the Hessians yields performance relatively close to the oracle, meaning that not much information is lost.
> 2. The Deadly Triad suggests that some settings, especially in the deep case, are bound to diverge. We did not rigorously test this, but such diverging settings seem to diverge, momentum or not, Adam or not. Instead what we find is that for settings that would converge with SGD (no-momentum), using momentum or Adam simply speeds up convergence. This was especially visible in Atari where learning rates can be an issue for convergence.
> 3. “maybe the authors can have the MSE plot for just Adam updates for a better comparison?”, perhaps we should make this clearer, in Figure 9 the Baseline line is just Adam. Or did you mean something else?
> 4. This is a fair criticism, we have added Figure 18 in the appendix for mountain car, and can add more. Extending the figures to include more steps simply shows gradual convergence over time.
> 5. This is also a fair criticism of no-target momentum methods, there is some inherent instability that (it seems) can only be resolved by decreasing learning rates or freezing the target network, but note that the X axis of Figure 17 spans 500k steps, 50 times the standard bootstrapping time of DQN. One thing we’ve found is that while our method is more precise in the short term, _it does not change the ordering of actions_, even compared to frozen-target DQN. That means that if an agent quickly identifies the optimal action it doesn’t matter how precise the underlying value estimation is, their performance in terms of reward will be the same.
>
> We believe our contribution is fundamentally to show that _deep_ RL methods are using momentum incorrectly, because the underlying assumptions of the supervised setting no longer hold. We find surprising the qualification of this work as incrementally building upon works such as that of Sun et al, since these are not the basis upon which this work is founded, as the linear framework of RL is quite limiting. Instead, we depart from Deep RL methods, pointing out a shared weakness, and providing a potential solution.

---

### Author Response · Authors · 2020-11-12
**Common response**

We thank the reviewers for their comments; it seems we need to work on the message and motivation we are trying to convey with our paper. We do believe that many concerns could simply be resolved both by a closer read on the reviewers’ part and by better writing on our part.

A shared concern is that of the scalability and applicability of the proposed method. This is a fair criticism, but we believe that a major contribution of this work is providing evidence of a previously unreported phenomenon. We do propose _a_ method to combat this phenomenon, but it would vain of us to claim victory over it; we certainly hope that future methods will be able to address this with better tools than Taylor expansions. Furthermore, this evidence pushes us to claim something deeper, which has been more or less ignored in recent literature, namely that “deep RL methods are not always best served by directly importing techniques from the supervised setting.”

We hope that reviewers will be able to appreciate our work through this perspective. We have individually responded to their concerns and are happy to engage in discussion for the remainder of the rebuttal period.

---

### Decision · Program_Chairs · 2021-01-07
**Final Decision**

**Decision:**

Reject

**Comment:**

This paper studies the role of momentum in temporal difference (TD) learning algorithms, and how this can be systematically exploited to accelerate the TD type algorithms. More specifically, the authors point out that the momentum term could be quite biased, and propose a scheme to remedy this issue. However, the reviewers point out the lack of motivation about bias correction; it is unclear why bias correction is crucial to achieve acceleration.